# Characterization of the Putative Acylated Cellulose Synthase Operon in *Komagataeibacter xylinus* E25

**DOI:** 10.3390/ijms23147851

**Published:** 2022-07-16

**Authors:** Izabela Szymczak, Agnieszka J. Pietrzyk-Brzezińska, Kajetan Duszyński, Małgorzata Ryngajłło

**Affiliations:** Institute of Molecular and Industrial Biotechnology, Lodz University of Technology, B. Stefanowskiego 2/22, 90-537 Lodz, Poland; izabela.redzynia@p.lodz.pl (I.S.); agnieszka.pietrzyk-brzezinska@p.lodz.pl (A.J.P.-B.); kajetan.duszynski@dokt.p.lodz.pl (K.D.)

**Keywords:** *Komagataeibacter*, bacterial cellulose, acylation, genome, transcriptome, 3D structure

## Abstract

Bacterial cellulose is a natural polymer with an expanding array of applications. Because of this, the main cellulose producers of the *Komagataeibacter* genus have been extensively studied with the aim to increase its synthesis or to customize its physicochemical features. Up to now, the genetic studies in *Komagataeibacter* have focused on the first cellulose synthase operon (*bcsI*) encoding the main enzyme complex. However, the role of other accessory cellulose operons has been understudied. Here we aimed to fill this gap by performing a detailed analysis of the second cellulose synthase operon (*bcsII*), which is putatively linked with cellulose acylation. In this study we harnessed the genome sequence, gene expression and protein structure information of *K. xylinus* E25 and other *Komagataeibacter* species to discuss the probable features of *bcsII* and the biochemical function of its main protein products. The results of our study support the previous hypothesis that *bcsII* is involved in the synthesis of the acylated polymer and expand it by presenting the evidence that it may also function in the regulation of its attachment to the cell surface and to the crystalline cellulose fibers.

## 1. Introduction

Bacterial nanocellulose (bionanocellulose, BNC) is a natural biopolymer built of β-1-4-linked glucose units, which has found numerous applications due to its outstanding structural, physical and mechanical features [1,2,3,4]. This polymer is naturally produced by many bacterial species as part of their biofilm. Some species produce high amounts of pure cellulose as their main exopolysaccharide. In particular, the strains of the *Komagataeibacter* (formerly *Gluconacetobacter*) genus have been widely recognized as the most efficient BNC producers [5,6,7]. Due to the high yield and the production of highly crystalline cellulose ribbons, the strains of this genus have often been selected as models for studying this process at the molecular, physiological and cell culture level [6,8,9,10,11]. Comparisons between the strains of this genus have shown that it is very diverse and that both the yield and properties of cellulose depend strongly on the genotype [7,12,13]. Since genome sequences of this genus accumulate, it became possible to explore the genetic basis of this phenotypic diversity. Answers to these questions are crucial for gaining a better control of the process of cellulose synthesis and its physicochemical features.

The central molecular machinery responsible for BNC biosynthesis consists of the enzyme complex of cellulose synthase (BCS). In the strains of the *Komagataeibacter* genus, the main enzyme complex is encoded by the operon, which consists of four core and three accessory genes (type I operon, *bcsI*). The core genes of the cellulose synthase complex are encoding β-glycosyltransferase catalyzing synthesis of β-1,4-glucan from UDP glucose (*bcsA*) and other subunits, which participate in β-glucan chain synthesis, translocation and packing of the fibrils at the cell surface (*bcsB, bcsC, bcsD*) [14]. The function of subunit BcsD is connected to the ability of the Bcs enzyme complex to produce highly crystalline cellulose [15]. The accessory genes: *cmcAx (endoglucanase)*, *ccpAx (*β*-glucosidase)* and *bglAx* (unknown function), are likely responsible for the modulation of the cellulose synthesis process [14,16]. The core and the accessory genes of the operon were shown in *K. medellinensis* ID13488 to be co-transcribed in one polycistronic mRNA [17]. A comparative genomics study has found that this operon is very well conserved in the genus with some variability in the *K. hansenii* strains (recently proposed to be reclassified to the *Novacetimonas* gen. nov.) [18], where the *bcsA* and *bcsB* subunits are fused [7]. Recently, a very detailed study by M. Bimmer et al. has confirmed that indeed, only the *bcsI* operon is responsible for the production of crystalline cellulose in *K. hansenii* ATCC 23760 [19].

Apart from the main *bcsI* operon, it was discovered that the *Komagataeibacter* genomes often harbor additional copies of the cellulose synthase operon [13,17,20,21]. The number of additional *bcs* operons vary between the genomes, but one that is always present encompasses four genes: *bcsAB*, *bcsX*, *bcsY* and *bcsCII*. This operon is commonly referred to as the second cellulose synthase operon (*bcsII*, type II); however, its function is much less characterized. Due to the sequence similarity of BcsY to a transacylase and BcsX to a deacylase, it is considered that *bcsII* is responsible for the production of the acylated, amorphous cellulose [22]. The presence of acylated cellulose has, however, never been confirmed. This operon has remained understudied probably because it is considered to have little influence on the total cellulose yield. Only recently have M. Bimmer et al. shed more light on the likely role of *bcsII* and shown that it is the product of an amorphous extracellular polymeric substance that covers and connects the bacterial cells [19]. The authors suggested that the role of the polymer produced by the second operon may be important for localization of the biofilm at the air–liquid interphase. The exact role of the proteins encoded by *bcsII* is yet to be characterized.

In this paper we focused first on a detailed analysis of the second cellulose synthase operon in *K. xylinus* E25. We further compared the sequence, structure and expression level of *bcsII* between other *Komagataeibacter* spp. by taking advantage of the available genomic sequences and the published transcriptomic profiles. Afterwards, we presented and discussed the predicted features and 3D structures of the proteins encoded by the *bcsII* operon. Finally, we showed the experimental results of solubility tests of *bcsII* proteins and of BcsX protein crystallization. The obtained results constitute a solid basis for further experimental investigation of the function of the proteins encoded by the *bcsII* operon.

## 2. Results and Discussion

### 2.1. Diversity of the BcsII Operon in the Genomes of the Komagataeibacter Genus

The genome of *K. xylinus* E25 harbors three *bcs* operons: *bcsI*, *bcsII* and *bcsIII* (Appendix A). The second *bcs* operon is of a similar size (13,301 bp) as the first operon and is predicted to consist of four, as already mentioned, genes (*bcsABII*, *bcsX*, *bcsY*, *bcsCII*) and an additional *bcsZ* gene, which was not described before (Figure 1 and Appendix A). The *bcsA* and *bcsB* subunits are fused in the case of both *bcsII* and *bcsIII* (Appendix A). Based on the annotation, the *bcsCII* gene is predicted to be disrupted by an insertion sequence of IS*1031*A-type, which has been described before in *K. hansenii* ATCC 23769 (former *Acetobacter xylinum*) and shown to be overrepresented among the *Komagataeibacter* species [23]. Due to the disruption by IS, the *bcsCII* is likely not to be functional in *K. xylinus* E25.

Next, we compared the sequence and the organization of the predicted *bcsII* operons of the type *Komagataeibacter* strains to understand how it is conserved in the genus (Figure 1). The second operon was present in all genomes; however, its structure displayed variability among the strains. Some of the differences are due to the incompleteness of the genome sequence (*K. intermedius* TF2 and *K. sucrofermentans* LMG 18788). The *bcsABII* gene is absent in *K. saccharivorans* LMG 1582; in *K. xylinus* LMG 1515 it is disrupted by a predicted IS66 family insertion sequence accessory protein TnpB and a transposase, which suggests that this operon is nonfunctional. Gene *bcsX* is the most highly conserved of all genes in the operon. More nucleotide sequence variability was discovered in the sequence of the *bcsY* gene, where in such strains as *K. diospyri* MSKU 9 and *K. oboediens* 174Bp2 it was slightly longer than in other strains, which translated into differences of 16 or 18 amino acids, respectively. This difference in *bcsY* size may, however, be due to incorrect prediction of the gene start. In the case of *bcsCII*, the highest sequence variability among the strains can be observed. This gene is disrupted or incomplete in *K. xylinus* E25, *K. medellinensis* NBRC 3288, *K. hansenii* ATCC 53582 and *Gluconacteobacter entanii* LTH 4560. Finally, gene *bcsZ* is much less conserved than *bcsX* or *bcsY*; in such strains as *K. diospyri* MSKU9, *K. rhaeticus* LMG 22126, *K. xylinus* CGMCC 2955, *K. hansenii* ATCC 53582 and *Ga. entanii* LTH 4560 it is absent. In *K. swingsii* LMG 22125 and *K. xylinus* E25 *bcsZ* was predicted to be slightly longer, which translated into a difference of 16 amino acids of the predicted protein. Similarly, as in the case of *bcsY*, this may be due to the incorrectly predicted gene start.

We decided to investigate in detail the cases of *bcsZ* absence by using tblastn. We found that that the genome sequence of *K. diospyri* MSKU 9 downstream of the *bcsII* produced significant matches with the translated BcsZ sequence, which suggests that the reading frame of *bcsZ* accumulated mutations during the strain’s evolution, which probably resulted in gene inactivation. In the case of the *K. intermedius* TF2 genome, which is highly fragmented, we found a significant match in another contig of the genome. For the genomes of *K. rhaeticus* LMG 22126 and *K. xylinus* CGMCC 2955 and the *K. hansenii* strains, however, no significant matches were produced. Given that the majority of *Komagataeibacter* spp. harbor *bcsZ*, it is very exceptional for the *K. rhaeticus* and *K. hansenii* strains to lack this gene.

By analyzing a wider genomic context of the *bcsII* operon, two distinguishable features may be observed for some of the *Komagataeibacter* species. In the case of *K. rhaeticus* LMG 22126, *K. xylinus* CGMCC 2955 and *K. medellinensis* NBRC 3288, *bcsII* is constituted of three additional genes (*kpsC*, *kpsS*, *rfaB*), which are located upstream of the *bcsABII* gene (Appendix A). This gene cluster may be involved in extracellular matrix formation, as described before for *K. medellinensis* ID13488 [17]. The presence of these three genes is unusual for the whole genus and is characteristic only to these two species, which have been previously shown to be phylogenetically closely related to each other [6].

Another unusual genomic context of *bcsII* can be observed for the strains of *K. saccharivorans*, which distinguishes them from other species of the *Komagataeibacter* genus (Appendix A). In these strains, *bcsII* starts immediately with *bcsX*, and is located upstream of the *bcsI* operon, which lacks the *bcsCI* gene. It can be assumed that the absence of *bcsCI*, which is crucial for the export of cellulose polymer, is compensated by the presence of *bcsCII*. What is more, although the type strain *K. saccharivorans* LMG 1582 does not produce cellulose (there is no phenotypic information for other sequenced strains), other strains of this species have been demonstrated to produce high yields of BNC [24,25]. It would be interesting to experimentally investigate how the structure of this unusual operon influences the properties of synthesized cellulose. The distinctiveness of the *K. saccharivorans* strains may be expected given the large phylogenetic distance from other *Komagataeibacter* strains as shown based on the genomic comparisons [6].

Finally, the genomic sequence of *K. oboediens* 174Bp2 is predicted to harbor four copies of the *bcsII* operon, which are located in different genomic contexts, as previously described [7]. How this high number of *bcsII* copies translates into the expression level of its proteins and further on the properties of cellulose polymer is yet to be investigated.

Summarizing, although the second cellulose operon is present in every *Komagataeibacter* genome, it displays much greater diversity than the first operon. The observed differences in the genomic context of *bcsII* suggest that it was independently gained during the evolution of the *Komagataeibacter* species and, therefore, its presence probably provided an adaptational advantage.

### 2.2. Expression Profile of the BcsII Operon

Apart from genome sequence data, recently, gene expression profiles have been published for *K. xylinus* E25 and *K. xylinus* CGMCC 2955 (probable *K. rhaeticus* sp.) [6] which opens the possibility to investigate in detail how cellulose operons are expressed [26,27].

The expression of genes of *bcsII* in *K. xylinus* E25, which is based on RNA-seq data, is shown in comparison with *bcsAI* of the first *bcs* operon (Figure 2a). It can be noticed that *bcsX* is the most highly expressed gene of *bcsII* and roughly three times higher than *bcsAI*. The expression of *bcsABII* and *bcsCII* is very low. *BcsY* and *bcsZ* are expressed at a much lower level than *bcsX.* Such variable expression of the *bcsII* genes suggests that, in the case of *K. xylinus* E25, they are regulated by a separate promoter or post-transcriptionally tuned. Additionally, we have observed that this operon is differentially regulated by the presence of ethanol in the culture medium (Appendix A). For the majority of *bcsII* genes this change is positive (for *bcsY* it is significant), as in the case of *bcsAI*, whereas *bcsZ* is significantly negatively regulated.

A slightly different expression profile of *bcsII* was captured in *K. xylinus* CGMCC 2955 (Figure 2b). Here, *bcsX* is the most highly expressed gene of the *bcsII* operon but at a much lower level, more than four times less than *bcsAI* of the first operon. The expression of *bcsABII* and *bcsCII* is the lowest of all genes of *bcsII.*

Additional insight into the expression of *bcsII* has been delivered by a study involving *K. medellinensis* ID13488, where it was shown that *bcsABII* was the most weakly expressed gene of all *bcs* operons present in the genome (the expression of other genes of *bcsII* was not measured) [17].

Taken together, the presented results suggest that the function of the *bcsII* operon is the synthesis of a modified cellulose, where *bcsX* may play the most important role. What is more, the presented profiles suggest that the expression of *bcsII* may differ in the *Komagataeibacter* spp. More research, however, is needed to investigate the dependence of genotype and condition on the expression of this operon.

### 2.3. Function Prediction, Sequence Similarity and Homology Search for the BcsII Proteins

We next focused on the detailed study of the unique proteins encoded by the *bcsII* operon of *K. xylinus* E25. First, a prediction of the functional domains within the protein sequences of the investigated proteins was made using the InterProScan database (Figure 3a). Nearly a whole sequence of BcsX is predicted to constitute the SGNH/GDSL hydrolase domain, whereas a sequence of the BcsY protein was predicted to carry the acyltransferase domain, as expected and described before [22]. This suggests that both proteins may play an important role during acylation of a newly-formed bacterial cellulose. Analysis of the BcsZ sequence showed that, similarly to BcsX, it contains the SGNH/GDSL hydrolase domain, but it is located near the C-terminus. Furthermore, these two proteins differ in size (223 vs. 576 amino acids) and share very low sequence similarity (local similarity of 30%). The function of the remaining part of BcsZ seems to be unclear; hypothetically, it may be involved in cooperation with the other protein partners in the BCS complex or may be responsible for allosteric regulation. The prediction of the SGNH/GDSL hydrolase domains in BcsX and BcsZ enhances the hypothesis that the *bcsII* operon is involved in the cellulose acylation process.

The sequence identity between BcsX, BcsY and BcsZ of *K. xylinus* E25 and their homologs of the *Komagataeibacter* species is high (100–75% for BcsX, 98–67% for BcsY and 97–57% for BcsZ; Appendix A). These results suggest that BcsX is the most highly conserved protein of all encoded by the *bcsII* operon. A search for homologues of BcsX and BcsZ in bacterial families other than *Acetobacteraceae* was also performed. The hydrolases of *Polynucleobacter aenigmaticus, Castellaniella defragrans* and *Pseudomonas alcaligenes* (Appendix A) were identical to BcsX of *K. xylinus* E25 in about 40%. The number of identical residues in a sequence was lower (25–30%) for the comparison of BcsZ with the homologues from non-*Acetobacteraceae* families (Appendix A). The top-ranked hydrolase homologues of BcsX and BcsZ were different, although both proteins belong to the same hydrolase family. This may be due to the fact that BcsZ contains an extra sequence proceeding the SGNH domain.

Interestingly, BcsX shares high sequence similarity (54%) with the WssF protein from *Pseudomonas fluorescens* SBW25, which produces cellulosic polymer consisting of partially acetylated cellulose that was found to provide an advantage for colonization of the air–liquid interface as well as rapid spreading across solid surfaces [28]. Studies of *wssF* and the adjacent downstream genes of *wssF* (*wssGHIJ*) mutants have shown that these genes are crucial for the synthesis of acetylated BC. WssF is a predicted GDSL-like lipase/acylhydrolase that was hypothesized to function by presenting acyl groups to the enzymes involved in the acetylation of cellulose. On the other hand, the adjacent downstream genes of *wssF* (*wssGHIJ*) of *P. fluorescens* SBW25, which are associated with the O-acetylation of alginate, are not present in *K. xylinus* E25. Despite the differences in operon structure, it is possible that BcsX has a similar role as WssF, but more studies are needed to explain its role and interaction with other proteins encoded by *bcsII* in the process of cellulose acetylation in *K. xylinus* E25.

The homologues of BcsY are widely distributed not only among the *Acetobacteraceae* family (*Rhodospirillales* order), but also among bacteria belonging to different orders, e.g., *Enterobacteriales*, *Rhizobiales* and *Pseudomonadales* (Appendix A and Appendix A). The sequence identity between BcsY and its homologues from different bacteria is variable (usually below 40%); however, the exception is a homologue from *Enterobacter lignolyticus*, which is identical with BcsY in 52%.

### 2.4. Prediction of BcsX, BcsY and BcsZ Subcellular Localization

Afterwards, we moved to the prediction of the subcellular localization and analysis of the secondary structure elements of the *bcsII* operon products.

Initially, we searched for the presence of a signal peptide (SP) in the sequences of the investigated proteins to predict their more probable localization in the cell. A signal peptide was predicted by SignalP only in the sequence of BcsZ (Appendix A). These results suggest that BcsZ functions outside of the cell.

Next, we used the Phobius program to predict the transmembrane (TM) topology of the *bcsII* proteins. Transmembrane α-helices were predicted only for BcsY (Figure 3e). The putative function of BcsY nicely correlates with the results of this analysis. Many bacterial acyltransferases are localized in the cytoplasmic membranes or lipophilic inclusions present in the cytosol [29]. Acyltransferases belong to the transferase enzyme class and catalyze the transfer of acyl groups from the donor to the acceptor, forming either esters or amides [30]. Sometimes the substrates are long-chain fatty acids; hence, it can be hypothesized that BcsY located in the cytoplasmic membrane might deal with such hydrophobic substrates available there. The secondary structure elements’ prediction of BcsY performed with PRED-TMBB2 (Appendix A) indicated that BcsY can be composed of fourteen transmembrane helices.

Additionally, Phobius predicted the presence of a signal peptide in the sequence of BcsX and BcsZ but not in BcsY. These predictions agree with those of SignalP in the case of BcsY and BcsZ (the same cleavage site predicted in both predictions); however, they differ for BcsX (Appendix A). The presence of a signal peptide in the sequence of BcsX is therefore questionable and so is its subcellular localization.

### 2.5. Three-Dimensional Structure Prediction for BcsX, BcsY and BcsZ

Recently, the models representing the predicted spatial structures of BcsX and BcsY were generated by AlphaFold and added to the corresponding UniProt entries (BcsX: Q9WX69, BcsY: Q9WX70). Additionally, we modelled the spatial structure of BcsZ using AlphaFold. As a next step, we performed structural homology searches for all proteins with the PDBeFold (SSM) server [31] using the AlphaFold models. BcsX and the C-terminal domain (CTD) of BcsZ (residues 393–576) show homology to many different serine esterases belonging to the SGNH/GDSL hydrolase family, whereas the middle parts of BcsZ (residues 192–382) represent a lectin-like fold. The five top-ranked homologs for each search are listed in Table 1.

The structural homology between BcsX and the listed proteins is relatively high. The superposition of BcsX on its homologs produced the rmsd values (calculated for Cα atoms) in the range from 2.31 to 2.67 Å (for 148–171 out of 262 Cα atoms of BcsX). Based on this structural comparison (Figure 3b), all α-helices and the central β-sheet are conserved among the discussed serine esterases, whereas several loop regions (residues 51–59, 78–84, 106–117, 141–152 and 178–186 in BcsX, residue numbering according to NCBI Gene: WP_025439176.1) have different conformations in the analyzed structures. Moreover, these loops also vary in length. An interesting element of the BcsX structure is an additional β-strand connected with a long loop (residues 51–77, shown in magenta in Figure 3b). All other structural homologues have a central β-sheet formed by five parallel β-strands; BcsX has this additional β-strand, which is, moreover, antiparallel to the adjacent β-strand.

The superposition of BcsZ-CTD on its homologs generated the rmsd values in the range 1.92–2.36 Å (for 147–159 out of 184 Cα atoms of BcsZ-CTD) indicating that the structural homology between BcsZ-CTD and its homologs is even higher than in the case of BcsX. In the model of BcsZ-CTD, all α-helices and the central part of the structure including the β-sheet are highly conserved (Figure 3c) and there are only three flexible loops (residues 470–479, 504–508 and 533–549, residue numbering according to NCBI Gene: WP_025439171.1). Interestingly, the rmsd value for superposition of BcsX and BcsZ-CTD is 2.75 Å (for 152 Cα atoms) and the SGNH/GDSL hydrolase domain of BcsX is much larger (262 residues) compared to BcsZ-CTD (184 residues). Hence, BcsX contains more variable regions.

The most important finding of the structural analysis of BcsX and BcsZ-CTD is the identification of the catalytic triad that involves Ser16, Asp187 and His190 in BcsX and Ser403, Asp550 and His553 in BcsZ-CTD located in conserved loop regions (Figure 3b,c, Appendix A). Another element important for the catalytic activity of serine esterases is an oxyanion hole. In the analyzed homological SGNH/GDSL hydrolases, it is usually formed by three residues, catalytic serine, glycine/alanine and asparagine. It is of note that in the case of serine and glycine/alanine residues, only their backbone atoms participate in the catalysis. In BcsX the corresponding residues are Ser16 and Ser46; however, if only the atoms of the residue backbone play a role, it does not matter that in BcsX there is serine instead of glycine/alanine. However, the third potential residue of the BcsX oxyanion hole is Thr104 (according to its position in the model). This might suggest that the mechanism of the BcsX catalysis can be slightly different. Another possibility is that the model requires improvement and another residue should be present at this position. For instance, closely located Asp103 could be a potential residue forming the oxyanion hole instead of Thr104. Interestingly, the oxyanion hole of BcsZ-CTD was easily identified, and it is formed by Ser403, Gly435 and Asn467 (Figure 3c and Appendix A). The conservation of the BcsZ-CTD sequence and spatial structure is high, and even the analysis performed by InterProscan indicated putative residues forming the catalytic triad and the oxyanion hole (the residue numbers were in agreement with the results of structural comparison). This finding confirms that BcsZ can act as the SGNH/GDSL hydrolase. Finally, we also compared the substrate-binding site of the analyzed homological enzymes, BcsX and BcsZ-CTD, but no significant similarities were detected. The residues interacting with substrates in other serine esterases were located mainly in the variable loop regions. Moreover, all analyzed enzymes utilize different substrates.

In addition to spatial structure analysis, we tried to pairwise compare the amino acid sequences of the mentioned proteins with the sequence of BcsX or BscZ-CTD using pairwise blastp. Interestingly, the only results were obtained for BcsX/*Mycobacterium smegmatis* serine esterase and BcsZ-CTD/*Escherichia coli* multifunctional enzyme. The sequence identity for the first pair was only 26% (residues 144–205 of BcsX, involving mainly conserved structural elements and two residues of the catalytic triad) and 25% for the latter one (residues 395–573 of BcsZ-CTD, covering the almost whole CTD). The comparison of amino acid sequences of BcsX and BcsZ indicates no significant similarity between these two proteins.

The search for structural homologs of the remaining part of BcsZ (residues 41–392 without the signal peptide) indicated that the middle part of the protein (residues 192–382) has a lectin-like fold (Figure 3d). The superposition of the BcsZ middle domain on its homologs produced the rmsd values in the range from 3.15–3.42 Å (for 149–165 out of 190 Cα atoms of the BcsZ middle domain). The top-ranked structural homologs are the L-type lectins. The members of this lectin family are usually present in the seeds of leguminous plants and their carbohydrate recognition domain is composed of two antiparallel β-sheets connected by short loops forming the “jelly-roll fold”. This structural motif can also be found in many other eukaryotic proteins [43] including the human N-terminal domain of thrombospondin that appeared in the results of the structural homologs search. The middle domain of BcsZ contains a lectin-like fold with two central β-sheets connected by a number of flexible loops of a different length. The most similar loop arrangement was observed for the BcsZ lectin-like domain and thrombospondin that mediates cell-to-cell and cell-to-matrix interactions, as it recognizes carbohydrate patterns [38]. However, the conformation of some β-strands forming the central β-sheets is also variable among the compared proteins (Figure 3d). Nevertheless, the obtained results suggest that this domain of BscZ might be involved in interacting with cellulose as it contains a fold that is usually involved in interactions with carbohydrates. The potential function of the N-terminal part of BcsZ (residues 41–191) remains unknown.

The results of the structural analysis of the BcsY model did not provide any information about conserved structural elements or the catalytic residues of this enzyme. The obtained rmsd values were high, except for two proteins, bovine cytochrome c oxidase subunit 3 (PDB ID: 3WG7, chain C) [44] and *Saccharomyces cerevisiae* translation initiation factor eIF-2B subunit delta (PDB ID: 6QG2, chain H) [45], the values of which were slightly below 4 Å (only over 108–130 out of 386 BcsY Cα atoms). Both proteins play different roles in eukaryotic organisms; hence, it is not very likely that they are structural homologues of BcsY, which is a bacterial protein and a putative acyltransferase. This classification is based only on conserved sequence fragments. To date, there is no information on the spatial structure of this acyltransferase family. The AlphaFold BcsY model contains 12 transmembrane α-helices (Figure 3f), whereas Phobius and PRED-TMBB2 predicted 11 and 14 transmembrane α-helices, respectively. The model contains large loop regions; hence, it is possible that the protein contains more helices than in the current model. The mechanisms of action of this enzyme, its substrates and products also remain unknown.

### 2.6. Overexpression and Solubility Tests of the BcsII Operon Proteins, BcsX Purification and Crystallization

To address the questions about the mechanisms of action of BcsX, BcsY and BcsZ, we tried to produce recombinant proteins that could be then used for structural studies. The expression and solubility of BcsX, BcsY and BcsZ were tested using *E. coli* BL21 Gold cells as the expression host. The *bcsX, bcsY* and *bcsZ* genes were cloned into expression vector pETM-11 and the proteins were bearing the N-terminal cleavable His_6_-tag. BcsX and BcsZ were detected in lysates obtained after the cultivation of bacteria transformed with the prepared constructs (Figure 4a). BcsY was not produced in *E. coli*; this might be related to the fact that BcsY is, according to our predictions, a transmembrane protein. Although the amount of BcsZ was higher compared to BcsX, only the latter protein was present in soluble protein fraction. Therefore, BcsX was subjected for further analysis.

Initially, BcsX was overproduced by bacteria growing in the LB medium, and the overexpression was triggered by IPTG. The 500 mL *E. coli* culture provided about 0.52 mg of purified protein. The use of autoinduction LB medium allowed the amount of protein to increase to 1.2 mg (per 500 mL culture).

BcsX purification was performed using a two-step protocol. After the first step of BcsX purification on Ni-NTA, the elution sample still contained some amount of other proteins (Figure 4b). The second purification step, including removal of the cleaved His_6_-tag, allowed the protein sample of higher purity to be obtained, although some weak bands corresponding to contaminants were observed on the SDS-PAGE gel (Figure 4c). Nevertheless, the obtained level of purity was sufficient for the initial crystallization trials. Small crystals of BcsX were obtained during the initial screening (Figure 5). The diffraction properties of these crystals were tested; however, the BcsX crystals were diffracted by X-ray only to 5 Å. Further crystallization optimization is necessary to obtain well-diffracting crystals, which can provide good-quality data sufficient for structure determination.

## 3. Materials and Methods

### 3.1. Bioinformatical Analysis of Genome and Transcriptome Data

Genome sequences of all *Komagataeibacter* strains with their annotation were downloaded from NCBI database (access date: December 2021). Alignments of *bcsII* operons were performed and visualized using CLC Sequence Viewer (v. 8.0).

Gene expression data (RNA sequencing, RNA-seq) for *K. xylinus* E25 came from [26]. The calculated FPKM (fragments per kilobase per million mapped fragments) values from 3 replicate libraries per condition were used (SH medium or SH medium + ethanol). RNA-seq data for *K. xylinus* CGMCC 2955 came from the study of [27]. Original NGS (next generation sequencing) libraries were downloaded from SRA (GSM5329218, GSM5329219, GSM5329220). The libraries represented 21% oxygen tension condition (used as reference in the original study), as stated by the authors. The sequencing reads were processed as described in [26]. The *K. xylinus* CGMCC 2955 (ASM276219v1) genome with annotation was downloaded from NCBI used during reads mapping. Gene expression data from the two studies was visualized using ggplot2 package [46] in R (v. 3.5.1).

### 3.2. Functional and Structural Prediction

Functional domain search in the protein sequences encoded by *bcsII* in *K. xylinus* E25 was performed using InterProScan web server of EBI [47]. Pairwise protein sequence alignment for *Komagataeibacter* strains was performed with DIAMOND in Blastp mode [48] and by using custom-made scripts written in Python 3.7.9. Search of homologs outside of the *Komagataeibacter* genus was performed using NCBI Blastp online server [49].

Prediction of signal peptides was conducted using SignalP 6.0 online service [50]. Phobius online service was used to predict transmembrane segments and, additionally, signal peptides [51,52]. The secondary structure elements prediction was performed with PRED-TMBB2 [53].

Three-dimensional structural models for BcsX and BcsY from *K. xylinus* NBRC 13693 generated by AlphaFold [54] were downloaded from UniProt (BcsX: Q9WX69, BcsY: Q9WX70) and visualized using PyMOL (Schrödinger, LLC, New York, NY, USA). The structure of BcsZ was predicted in this study using AlphaFold v. 2.1.1. Structural homology searches were performed with the PDBeFold (SSM) server [31].

### 3.3. Overexpression and Solubility Tests of the BcsII Operon Proteins

The *bcsX, bcsY* and *bcsZ* genes were PCR-amplified from a genomic DNA of *Komagateibacter xylinus* E25 (isolated as described before in [7]) and inserted via NcoI and EcoRI (*bcsX*) or MunI and XhoI (*bcsY, bcsZ*) restriction sites into the pETM-11 vector (EMBL Heidelberg) under the control of a T7 promotor for production of fusion protein bearing a TEV-cleavable N-terminal His_6_-tag. The primers used in PCR are listed in Appendix A.

*E. coli* BL21 Gold competent cells were transformed with the prepared vectors. Bacteria were cultivated in lysogeny broth (LB) medium (50 mL culture per construct, supplemented with 50 μg/mL of kanamycin) at 37 °C to an OD_600_ of ~0.6. Subsequently, isopropyl-β-D-thiogalactopyranoside (IPTG) was added to final concentration of 0.5 mM. After next 3 h of overexpression at 37 °C, the cells were pelleted. The cell pellets were resuspended either in buffer A (20 mM HEPES, pH 7.5, 500 mM NaCl, 10 mM imidazole, 1 mM DTT supplemented with 0.1 mM PMSF) or buffer B (20 mM HEPES, pH 7.5, 500 mM NaCl, 10 mM imidazole, 10% glycerol, 1 mM DTT supplemented with 0.1 mM PMSF) and lysed by sonication using a Sonopuls GM 3200 ultrasonic homogenizer (Bandelin). Lysates were centrifuged for 40 min at 4 °C and 10,000× *g*. The samples representing lysates (collected prior centrifugation) and soluble protein fraction (collected after centrifugation) were analysed using SDS-PAGE.

### 3.4. Production and Purification of BcsX Protein

Initially, overproduction of BcsX was carried out as described above. In total culture volume of 500 mL. Subsequently, to increase the amount of produced protein, autoinducing medium, LB Broth Base including trace elements (Formedium), was used and bacteria were cultivated at 37 °C to an OD_600_ of ~0.6 and subsequently further incubated at 20 °C for 18 h. The cells were harvested at an OD_600_ of ~10.

BcsX purification protocol was based on the previously described purification of other recombinant proteins [55,56]. The cell extracts were prepared as described in Section 3.3. The target protein was captured from the cleared lysate on 0.5 mL of Ni-NTA agarose beads (Qiagen) and eluted with 400 mM imidazole. The His_6_-tag was cleaved with TEV protease during overnight dialysis at 4 °C against 20 mM HEPES, pH 7.5, 500 mM NaCl, 10% glycerol, 1 mM DTT. The cleaved protein was loaded onto 0.5 mL of Ni-NTA agarose to remove the His_6_-tagged TEV protease, uncleaved protein and cleaved His_6_-tag. The flow-through fractions were concentrated to 11.4 mg/mL.

### 3.5. Protein Crystallization and Data Collection

Initial screening for crystallization conditions was performed at 18 °C using Index Screen, SaltRx1 Screen and PEG/Ion Screen (Hampton Research) and the hanging-drop vapor diffusion method with drops containing 1 µL of protein solution and 1 µL of reservoir solution. The initial crystals of BcsX were obtained using the following solutions: (1) 2.1 M DL-Malic acid pH 7.0; (2) 0.2 M sodium chloride, 0.1 M Tris pH 8.5, 25% PEG 3350; (3) 0.2 M ammonium acetate, 0.1 M HEPES pH 7.0, 25% PEG 3350. Crystals obtained using solutions 2 and 3 were cryoprotected by transfer into mother liquor containing 22.5 % (*v/v*) PEG 400 and flash-cooled in liquid nitrogen. X-ray diffraction measurements were carried out on beamline 14.1 of the BESSY II storage ring (Berlin, Germany; [57]).

## 4. Conclusions

In comparison to the first cellulose synthase operon, the role of *bcsII* has not been extensively studied and only putatively assigned to the synthesis of amorphous, acylated cellulose. The evidence supporting this hypothesis was, however, lacking.

In this study we aimed to gain a deeper insight into the function of *bcsII*. First, by taking the advantage of the accumulated genomic sequences of the genus, we compared the conservation of this operon. We found that this operon is present in every *Komagataeibacter* genome; however, its structure differs depending on the species. We discovered that in *K. rhaeticus* and *K. medellinensis*, this operon is associated with *kps* genes, which are involved in extracellular matrix formation. Association with these genes supports the observation of M. Bimmer et al. that *bcsII* is responsible for the synthesis of polymer that is attached to the bacterial cell. Further sequence similarity analyses have shown that the proteins encoded by *bcsII* are quite unique to the *Komagataeibacter* genus. For BcsX, relatively high sequence similarity was found with the WssF protein of *P. fluorescens* SBW25, which was confirmed to be important for the synthesis of acetylated cellulose. Our study revealed for the first time the presence of another gene, *bcsZ*, which directly follows *bcsII*.

We further found that the expression of *bcsX* and *bcsY*, unlike that of *bcsZ*, was positively regulated in the medium containing ethanol, which are the conditions promoting cellulose biosynthesis in *K. xylinus* E25. This can suggest that there may be a positive regulatory connection between the cellulose biosynthesis pathway and the expression of *bcsX* and *bcsY*. The opposite seems to be the case for *bcsZ*; however, more expression profiles are needed to confirm the coupling of *bcsII* genes’ expression with cellulose synthesis.

By focusing on the protein products of the *bcsII* operon we found that both BcsX and BcsZ contain domains belonging to the SGNH/GDSL family, which is characteristic to serine esterases. Their pairwise sequence similarity was, however, low, which suggests that these enzymes may play different roles. Based on the homology search of BcsZ’s predicted 3D structure, we found that, apart from the SGNH/GDSL domain, the middle part of this protein has a lectin-like fold that is similar to human thrombospondin, which mediates cell-to-cell and cell-to-matrix interactions by recognizing carbohydrate patterns. We therefore postulate that BcsZ may play a similar role in the attachment of *Komagataeibacter* cells to the acylated and the crystalline cellulose. The prediction of the signal peptide at the N-terminus of BcsZ further suggests that this enzyme functions outside of the cell.

In summary, based on the gathered results, we postulate that the role of *bcsII* is not only the synthesis of the acylated polymer but also regulation of its attachment to the cell surface and to the crystalline cellulose fibers. As suggested previously for this operon, its function may influence colonization of the air–liquid interphase, and, through analogy to WssF function, spreading across solid surfaces. These findings, together with the experimental results of the *bcsII* operon proteins’ purification and crystallization, constitute a solid base for further studies on their exact function.

## Figures and Tables

**Figure 1 ijms-23-07851-f001:**
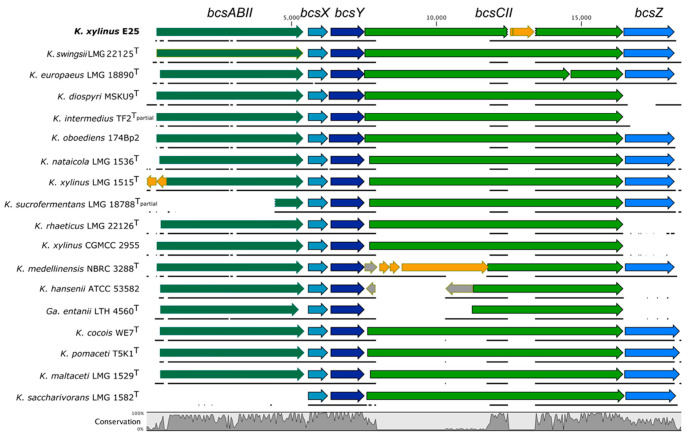
Comparison of *bcsII* operons in the type *Komagataeibacter* strains. For *K. intermedius* TF2, *K. sucrofermentans* LMG 18788, only a partial operon sequence was enclosed within a draft assembly. Additional (not type) strains of the same species were included where the type strain’s genome assembly was fragmented, or a strain was important for the study. Genes colored in orange represent predicted IS elements or transposases. ^T^ stands for the type strain.

**Figure 2 ijms-23-07851-f002:**
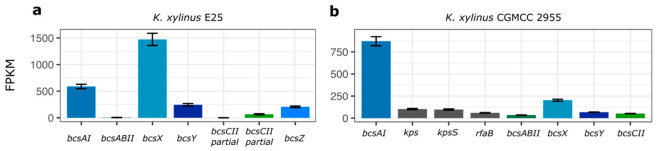
Comparison of gene expression profile of the *bcsII* operon and *bcsAI* based on published RNA-seq data. (**a**) *K. xylinus* E25. (**b**) *K. xylinus* CGMCC 2955. Presented values are the mean calculated from 3 replicates. Transcripts mean FPKM values of 3 replicates are shown. Thin black bars denote standard error.

**Figure 3 ijms-23-07851-f003:**
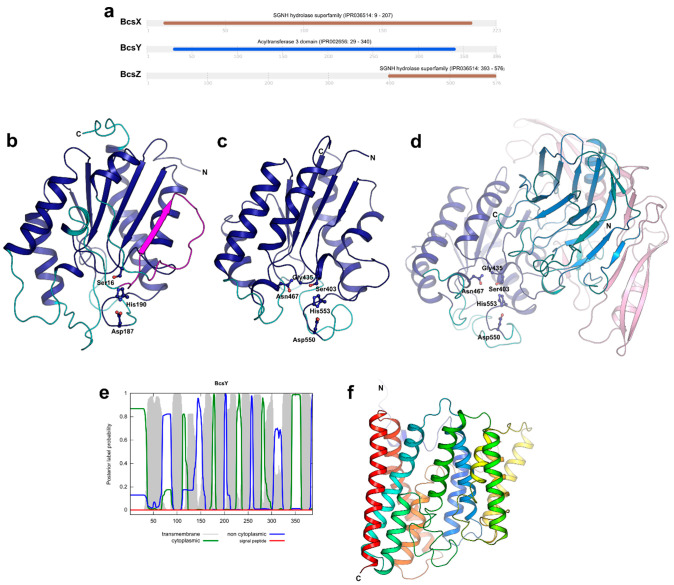
Predicted features of *bcsII* proteins. (**a**) Prediction of functional domains for the amino acid sequences of BcsX, BcsY and BcsZ by InterProScan. (**b**) The predicted spatial structure of BcsX. The conserved structural elements and flexible loops of BcsX are shown in blue and cyan, respectively. The structural element unique for BcsX is presented in magenta. The side chains of the catalytic triad residues are shown as sticks and spheres. (**c**) The predicted spatial structure of BcsZ-CTD. The conserved structural elements and flexible loops of BcsX are shown in blue and cyan, respectively. The side chains of the catalytic triad residues are shown as sticks and spheres. (**d**) The predicted spatial structure of BcsZ. The domains were presented in different colors: NTD—pink, lectin-like domain—marine/teal (conserved structural elements/variable conformations), CTD—blue/cyan (conserved structural elements/flexible loops). (**e**) Predicted transmembrane topology of BcsY. The analysis was performed using Phobius. (**f**) The model of BcsY is rainbow-colored to visualize the twelve transmembrane helices.

**Figure 4 ijms-23-07851-f004:**
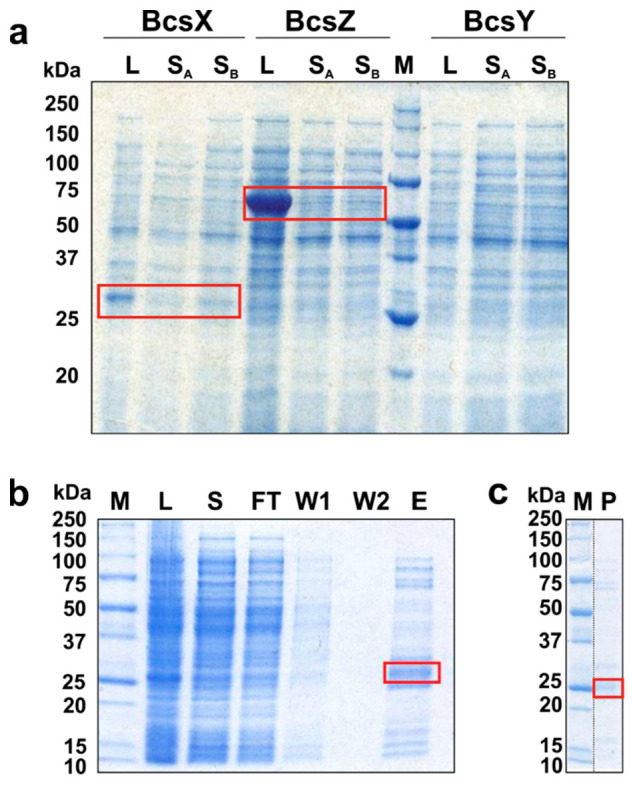
Overexpression and purification of *bcsII* proteins. (**a**) SDS-PAGE electrophoregram of the lysate and supernatant samples obtained after protein production in *E. coli* expression system (M—protein molecular weight marker, L—lysate, S_A_ and S_B_—supernatant samples obtained with either buffer A or B, respectively). (**b**). SDS-PAGE results of the first step of BcsX purification on Ni-NTA (M—protein molecular weight marker, L—lysate, S—supernatant, FT—flow-through fraction, W1 and W2—wash fractions, E—elution fraction). (**c**). SDS-PAGE results of the BcsX purification, P represents the final sample of BcsX obtained after purification and used for crystallization. The bands corresponding to the studied proteins are marked with red boxes.

**Figure 5 ijms-23-07851-f005:**
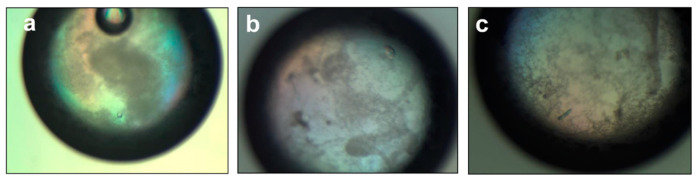
BcsX crystals. Initial crystals of BcsX were obtained using the following crystallization solutions: (**a**) 2.1 M DL-Malic acid pH 7.0; (**b**) 0.2 M sodium chloride, 0.1 M Tris pH 8.5, 25% PEG 3350; (**c**) 0.2 M ammonium acetate, 0.1 M HEPES pH 7.0, 25% PEG 3350. Microscope magnification: 50×.

**Table 1 ijms-23-07851-t001:** Structural homologs of BcsX and BcsZ proteins.

PDB ID	Enzyme	Rmsd (Å)	Cα Atoms Aligned	Reference (Homolog Structure)
** *Structural homologs of BcsX* **				
4TX1	*Sinorhizobium meliloti* carbohydrate acetylesterase	2.63	171	[32]
3DCI	*Agrobacterium tumefaciens* putative arylesterase	2.67	168	n/a
2Q0Q	*Mycobacterium smegmatis* serine esterase	2.47	160	[33]
4JGG	*Pseudomonas aeruginosa* serine esterase	2.57	154	[34]
3HP4	*Pseudoalteromonas sp*. 643A serine esterase	2.31	148	[35]
** *Structural homologs of BcsZ-CTD* **				
1IVN	*Escherichia coli* multifunctional enzyme (activities of thioesterase, esterase, arylesterase, protease and lysophospholipase)	1.92	147	[36]
3P94	*Parabacteroides distasonis* GDSL-like lipase	1.94	151	n/a
5B5S	*Talaromyces cellulolyticus* carbohydrate esterase	2.01	152	[37]
4TX1	*Sinorhizobium meliloti* carbohydrate acetylesterase	2.36	159	[32]
2Q0Q	*Mycobacterium smegmatis* serine esterase	2.10	151	[33]
** *Structural homologs of BcsZ middle part (residues 192–382)* **				
1Z78	*Homo sapiens* thrombospondin-NTD	3.21	163	[38]
3IPV	*Spatholobus parviflorus* seed lectin	3.39	165	[39]
1LU1	*Vigna unguiculata* seed lectin	3.42	161	[40]
1FX5	*Ulex europaeus* lectin I	3.15	149	[41]
1GSL	*Griffonia simplicifolia* lectin	3.24	150	[42]

## Data Availability

Not applicable.

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
