# Peer review of "Characterization of the Putative Acylated Cellulose Synthase Operon in Komagataeibacter xylinus E25"

_ijms, 2022, doi:10.3390/ijms23147851_

Round 1

Reviewer 1 Report

Authors present work on the bacterial gene pathways regulating production of cellulose. The study mostly operates with the data acquired from the open databases, meaning that we are finally getting in the stage of analyzing huge quantities of genome sequences, which were accumulated for the past couple of decades. The results are clearly presented and are easy to understand for the reader outside the field. All the figures in the main text are good, but I don’t know if the tables with sequence identities are necessary there. Manuscript is written in a concise language, only a minor spell check is required. Experiment is comprehensible, but I wonder if authors plan to knock off genes from bcsII operon to confirm predicted function. In other aspects everything else is fine, I have only some suggestions for spot corrections:

Line 105 – how did you choose genomes of Komagataeibacter strains? In one place you specify, that they were type strains, but in the other that there also were non-type strains “important for the study”. What does it mean? Maybe you could add supplementary table with specifics of the strains used in the study?

Line 268 – what’s the point in mentioning that these taxa represent Gram-negative bacteria?

Line 460 – phrase «with annotation» is mentioned twice in a sentence

Line 470 – did access date was really in June 2020, and not 2022?

Line 478 –phrase “The structure o BcsZ” is missing “f”

Lines 494-496 and 508-510 contain the same information.

Lines 528-533. Usually, conclusions cover only findings in the original research and do not cover literature data. So, I advise to skip this part.

Line 537, 556 (and some other places) I find that words “interesting” and “important” do not usually add value to the point. So, I advise to remove “one interesting finding” and just leave “we discovered”

Line 555. I would change the sentense to “Their pairwise sequence similarity was, however, low which suggests that these enzymes may play different roles

Author Response

Authors present work on the bacterial gene pathways regulating production of cellulose. The study mostly operates with the data acquired from the open databases, meaning that we are finally getting in the stage of analyzing huge quantities of genome sequences, which were accumulated for the past couple of decades. The results are clearly presented and are easy to understand for the reader outside the field.

We thank the reviewer for considering our work as clearly presented and easy to understand. We have addressed the reviewer’s remarks as detailed below.

All the figures in the main text are good, but I don’t know if the tables with sequence identities are necessary there.

According to reviewer’s suggestion, we transferred the mentioned table to Supplemetary Materials.

Manuscript is written in a concise language, only a minor spell check is required. Experiment is comprehensible, but I wonder if authors plan to knock off genes from bcsII operon to confirm predicted function.

Knock out of the bscII operon genes is our next research goal. However, to perform this experiment some more time is required and the obtained results will be included in the next article concerning this operon.

In other aspects everything else is fine, I have only some suggestions for spot corrections:

Line 105 – how did you choose genomes of Komagataeibacter strains? In one place you specify, that they were type strains, but in the other that there also were non-type strains “important for the study”. What does it mean? Maybe you could add supplementary table with specifics of the strains used in the study?

We thank the reviewer for these comments and for pointing out the imprecise description of the chosen strains. Indeed, we used the genomes of the type, representative of the species, strains of the Komagataeibacter and denoted them with T superscript in the strains names in Figure 1. However, it turned out that some of the genomes are of low quality and needed to be replaced with another (not type) strain of the same species. Furthermore, we based our analyses, throughout the manuscript, on some not type strains like K. Xylinius E25 or K. xylinius CGMCC 2955 and thought that it would be sensible to add these additional strains to Figure 1. We added the following description to Figure 1: “T stands for the type strain”.

Line 268 – what’s the point in mentioning that these taxa represent Gram-negative bacteria?

We have removed the statement: “The common feature of the mentioned bacteria is that they are all classified as Gram-negative".

Line 460 – phrase «with annotation» is mentioned twice in a sentence

We thank the reviewer for pointing this mistake. We have corrected this sentence accordingly.

Line 470 – did access date was really in June 2020, and not 2022?

We thank the reviewer for pointing this mistake, it should be 2022.

Line 478 –phrase “The structure o BcsZ” is missing “f”

Corrected.

Lines 494-496 and 508-510 contain the same information.

We have changed the repetitive statement to: “The cell extracts were prepared as described in Section 3.3”.

Lines 528-533. Usually, conclusions cover only findings in the original research and do not cover literature data. So, I advise to skip this part.

According to the reviewer’s suggestion, we have removed the statement: “Recently, it has been shown that the extracellular polymer produced by bcsII in K. hansenii ATCC 23769 is coating and connecting the bacterial cells and probably also the crystalline cellulose fibers (Bimmer et al. 2022)”.

Line 537, 556 (and some other places) I find that words “interesting” and “important” do not usually add value to the point. So, I advise to remove “one interesting finding” and just leave “we discovered”

According to the reviewer’s suggestion, we have changed the mentioned sequence.

Line 555. I would change the sentense to “Their pairwise sequence similarity was, however, low which suggests that these enzymes may play different roles”

Corrected.

Reviewer 2 Report

This study aims at the analysis of the second cellulose synthase operon in K. xylinus E25. Furthermore, the authors compared the sequence, structure, and expression level of the bcsII between other Komagataeibacter spp. by taking the advantage of the available genomic sequences and the published transcriptomic profiles. Moreover, the predicted features and 3D structures of the proteins encoded by the bcsII operon were also discussed. Finally, the authors also showed the experimental results of solubility tests of bcsII proteins and BcsX protein crystallization. Therefore, it can be summarized, that the manuscript constitutes a consistent work for further experimental investigation of the function of the proteins encoded by the bcsII operon.

The paper is written within the Journal required style. The whole study was seriously conducted and methodology applied is scientifically sound. The manuscript is well-designed, results and discussion are plausible and coherent according to the binding scientific standards; they are also explicatory and sufficiently presented with several well-prepared figures and tables. The objective of the study is well defined and the Authors explain how they reached the conclusions. The paper is also written correctly in terms of its language and in my opinion, it is suitable for publication in its present form.

The only comment I have concerns the last part of the work. The authors summarize their work by formulating a hypothesis. However, in my opinion, the hypothesis should be rather formulated at the end of the Introduction chapter and should precede the main aim of the study. Therefore, I suggest re-editing the last paragraph of the work (L560 – 570) so that it refers to the conclusion resulting from the obtained results, and does not suggest that it is a hypothesis.

Author Response

This study aims at the analysis of the second cellulose synthase operon in K. xylinus E25. Furthermore, the authors compared the sequence, structure, and expression level of the bcsII between other Komagataeibacter spp. by taking the advantage of the available genomic sequences and the published transcriptomic profiles. Moreover, the predicted features and 3D structures of the proteins encoded by the bcsII operon were also discussed. Finally, the authors also showed the experimental results of solubility tests of bcsII proteins and BcsX protein crystallization. Therefore, it can be summarized, that the manuscript constitutes a consistent work for further experimental investigation of the function of the proteins encoded by the bcsII operon.

The paper is written within the Journal required style. The whole study was seriously conducted and methodology applied is scientifically sound. The manuscript is well-designed, results and discussion are plausible and coherent according to the binding scientific standards; they are also explicatory and sufficiently presented with several well-prepared figures and tables. The objective of the study is well defined and the Authors explain how they reached the conclusions. The paper is also written correctly in terms of its language and in my opinion, it is suitable for publication in its present form.

We thank the reviewer for considering our work as well-designed and well-written. We have addressed the reviewer’s comment as detailed below.

The only comment I have concerns the last part of the work. The authors summarize their work by formulating a hypothesis. However, in my opinion, the hypothesis should be rather formulated at the end of the Introduction chapter and should precede the main aim of the study. Therefore, I suggest re-editing the last paragraph of the work (L560 – 570) so that it refers to the conclusion resulting from the obtained results, and does not suggest that it is a hypothesis.

According to the reviewer’s suggestion, we have changed the phrases “hypothesize” to “postulate” and the phrase “To summarize our hypotheses” to “In summary”.